# The Polish Society of Gynecological Oncology Guidelines for the Diagnosis and Treatment of Endometrial Carcinoma (2023)

**DOI:** 10.3390/jcm12041480

**Published:** 2023-02-13

**Authors:** Jacek J. Sznurkowski, Janusz Rys, Artur Kowalik, Agnieszka Zolciak-Siwinska, Lubomir Bodnar, Anita Chudecka-Glaz, Pawel Blecharz, Aleksandra Zielinska, Andrzej Marszalek, Mariusz Bidzinski, Wlodzimierz Sawicki

**Affiliations:** 1Department of Surgical Oncology, Medical University of Gdansk, ul. Smoluchowskiego 17, 80-214 Gdańsk, Poland; 2Department of Tumor Pathology, Maria Sklodowska-Curie National Research Institute of Oncology, Kraków Branch, 44-102 Kraków, Poland; 3Department of Molecular Diagnostics, Holy Cross Cancer Center, 25-735 Kielce, Poland; 4Division of Medical Biology, Institute of Biology, Jan Kochanowski University, 25-406 Kielce, Poland; 5Department of Gynaecological Oncology, Maria Sklodowska-Curie National Research Institute of Oncology, 00-001 Warsaw, Poland; 6Faculty of Medical and Health Sciences, Siedlce University of Natural Sciences and Humanities, 08-110 Siedlce, Poland; 7Department of Gynecological Surgery and Gynecological Oncology of Adults and Adolescents, Pomeranian Medical University, 70-111 Szczecin, Poland; 8Department of Gynaecologic Oncology, Maria Sklodowska-Curie National Research Institute of Oncology, Krakow Branch, 44-102 Kraków, Poland; 9Department of Obstetrics and Gynecological Oncology, Medical University of Warsaw, 00-575 Warsaw, Poland; 10Department of Tumor Pathology and Prophylaxis, Poznan University of Medical Sciences, 61-701 Poznan, Poland; 11Department of Tumor Pathology, Greater Poland Cancer Center in Poznan, 61-866 Poznan, Poland

**Keywords:** endometrial cancer, guidelines, AGREE, POLE, MMRD, NSMP, TP53abn, immunotherapy, adjuvant, follow-up, hereditary cancer

## Abstract

Background: Due to the increasing amount of published data suggesting that endometrial carcinoma is a heterogenic entity with possible different treatment sequences and post-treatment follow-up, the Polish Society of Gynecological Oncology (PSGO) has developed new guidelines. Aim: to summarize the current evidence for diagnosis, treatment, and follow-up of endometrial carcinoma and to provide evidence-based recommendations for clinical practice. Methods: The guidelines have been developed according to standards set by the guideline evaluation tool AGREE II (Appraisal of Guidelines for Research and Evaluation). The strength of scientific evidence has been defined in agreement with The Agency for Health Technology Assessment and Tariff System (AOTMiT) guidelines for scientific evidence classification. The grades of recommendation have been based on the strength of evidence and the level of consensus of the PSGO development group. Conclusion: Based on current evidence, both the implementation of the molecular classification of endometrial cancer patients at the beginning of the treatment sequence and the extension of the final postoperative pathological report of additional biomarkers are needed to optimize and improve treatment results as well as to pave the route for future clinical trials on targeted therapies.

## 1. Background

The Polish Society of Gynecological Oncology (PSGO) has developed the following recommendations for diagnosis, preoperative assessment for surgical treatment, radiotherapy, systemic treatment, treatment of recurrent disease and post-treatment surveillance of patients with endometrial carcinoma according to standards set by a guideline evaluation tool Appraisal of Guidelines for Research and Evaluation (AGREE) II [1].

Recommendations apply to women over the age of 18 who suffer from endometrial carcinoma (they do not apply to patients with other malignant neoplasms of the uterus, e.g., of the cervix and uterine sarcomas).

The recommendations are intended for gynaecologists, gynaecological oncologists, surgeons, pathologists, geneticists, radiotherapists, clinical oncologists, general practitioners, palliative care specialists and allied health care professionals.

The PSGO guidelines were developed using a six-step process:Nomination of a multidisciplinary development group (gynaecological oncologist/pathologist/geneticist/clinical oncologist/radiation oncologist).Identification of scientific evidence.Formulation of guidelines.Assessment of coherence with ESGO/NCCN guidelines.Evaluation of guidelines by external reviewers.Integration of external reviewers’ comments with the original content of the guidelines.

The strength of scientific evidence was defined in agreement with The Agency for Health Technology Assessment and Tariff System (AOTMiT) guidelines for scientific evidence classification [2] (Table 1).

The grades of recommendation were based on the strength of evidence and level of consensus of the PTGO development group as described in Table 2.

Any clinician intending to apply PSGO guidelines is expected to perform a careful medical evaluation of individual clinical circumstances to determine the best course of patient care and/or treatment.

For the first time, recommendations were developed according to standards set by the guideline tool Appraisal of Guidelines for Research and Evaluation (AGREE) II in gynaecologic oncology.

The strengths of the guidelines are they are comprehensive and up-to-date.

The limitation is the uncertainty of how many of the authors using guideline AGREE II were methodological experts.

Recommendations indicate an urgent need for changes in the system of financing health services in Poland.

## 2. Classification of Endometrial Carcinoma

### 2.1. Histopathological

The 2005 WHO classification (5th edition) identifies 10 histological subtypes of endometrial cancer (Table 3) [3,4,5,6,7,8] (strength of evidence V).

In terms of pathogenesis, endometrial cancer was categorised by Bokhman into two subgroups:type I—endometrioid carcinoma (adenocarcinoma) (80–90% of all diagnoses).type II—non-endometrioid carcinoma, which includes serous carcinoma, clear cell adenocarcinoma, undifferentiated and dedifferentiated carcinoma, mixed carcinoma of the endometrium and carcinosarcoma [9] (strength of evidence V).

### 2.2. Molecular

The latest morphological and clinical studies have shown that both the traditional histological classification [3,4,5,6,7,8] and the two pathogenic types of endometrial carcinoma according to Bokhman [9] do not allow for reliable assessment of prognosis and response to treatment. Obtaining this type of information is possible thanks to the molecular classification of endometrial cancer introduced in 2013 as The Cancer Genome Atlas (TCGA) sponsored by the National Cancer Institute (NCI) and by The National Human Genome Research Institute. The aforementioned classification identifies four molecular subtypes of endometrial cancer—POLE, MMRd/MSI-H, TP53-mutated (abn) and TP53wt-NSMP—differing in mutation profile, immunogenicity and prognosis [10] (strength of evidence IVA) and requiring different management [10,11,12] (strength of evidence IVD, V and V) (Table 4).

## 3. Diagnosis

### 3.1. Endometrial Biopsy

The biopsy of the endometrium (abrasion, aspiration biopsy and hysteroscopic biopsy) is recommended for

(1)Women with postmenopausal bleeding whose endometrial thickness is > 3 mm [13] (strength of evidence IIIA) (grade of recommendation 2A);(2)Women with adult granulosa cell tumour undergoing fertility-sparing treatment (with preservation of the uterus) [14] (strength of evidence IVA) (grade of recommendation 2B);(3)Women with bleeding during tamoxifen treatment lasting up to 5 years [15] (strength of evidence IVA) (grade of recommendation 2B).

There are no data regarding the safety of such an approach for patients using tamoxifen for up to 10 years. Tamoxifen therapy beyond 5 years significantly increases the risk of endometrial cancer (HR-1.74) [16] (strength of evidence IIA). Caution is recommended in this group of women (transvaginal ultrasound evaluation of the uterus every 6 months) (expert opinion) (strength of evidence V) (grade of recommendation 2B).

Regardless of the duration of tamoxifen use, special attention should be paid to menopausal women who can be asymptomatic due to stenosis of the cervical canal (grade of recommendation 1).

The endometrial thickness cut-off at which biopsy in asymptomatic women in the general population (pre- and postmenopausal) and in those with increased risk of endometrial cancer (PCOS, obesity, no childbirths or late menopause) *, which would have acceptable sensitivity and specificity, has not been established. Thus, an individual approach is recommended (expert opinion) (strength of evidence V) (grade of recommendation 2B). * Note: The management of endometrial hyperplasia (a precancerous condition) is a separate subject not covered by this recommendation.

The sensitivity of endometrial biopsy (cumulative value for abrasion and aspiration biopsy) is 89% and the false-negative rate is 10% [17] (strength of evidence IVA) [18] (strength of evidence IIIA).

The sensitivity of an adequate aspiration biopsy is significantly higher: 91% for premenopausal and 99.6% for postmenopausal women [19] (strength of evidence IIIA). However, obtaining tissue material adequate for the histopathological assessment using this method concerns 85% of samples [18] (strength of evidence IIIA).

For the reasons mentioned above, there is no preferred method of endometrial biopsy (grade of recommendation 2A).

### 3.2. Histopathological Report of Endometrial Biopsy

The report should specify the histological type, and for endometroid carcinoma, its differentiation grade assessed in two categories: low-grade (G1/G2) and high-grade (G3) [3,4,5,6,7,8] (strength of evidence V) (grade of recommendation 2B).

The existing scientific evidence indicates

(1)The advantage of the new molecular classification over the former based solely on the type of endometrial cancer and grading in making therapeutic decisions at the beginning of treatment [20,21,22] (strength of evidence IIIE, IIIA and IIA);(2)A significantly higher concordance between pre- and postoperative results for the new molecular classification based on the ProMisE classifier and/or sequencing compared to previously considered features (type and grading) [23,24,25,26,27,28] (strength of evidence IIIB, IIIB, IIID, IIID, IIID, IIIB),(3)High sensitivity and specificity of the ProMisE classifier [29,30] (strength of evidence IIID and IIIC), which is potentially realisable in most pathomorphology units. It is recommended that molecular classification be defined (at least a basic variant of ProMisE) at the initial diagnosis of endometrial cancer (biopsy), and if this is impossible, it should be performed at the latest before the decision on adjuvant treatment (grade of recommendation 2A).

CAUTION: Every woman with endometrial cancer for whom fertility-sparing treatment is being considered must obligatorily be subject to molecular classification (at least a basic variant of ProMisE). A similar requirement applies to high-risk patients with comorbidities who do not qualify for surgical treatment (grade of recommendation 2A).

A detailed description of the ProMisE classifier and comprehensive endometrial carcinoma diagnosis algorithm NGS+IHC is included in Appendix A.

The centres where the diagnostic minimum (ProMisE molecular classification) cannot be performed may, in the transitional period, use existing criteria: type and grading (does not apply to the decision on fertility-sparing treatment and management of nonoperable cases) (grade of recommendation 3).

### 3.3. Final Postoperative Pathological Report (Examination of the Uterus)

The report should include a verified histological type, and for endometroid carcinoma, verified grading assessed in two categories: low-grade (G1/G2) and high-grade (G3) [3,4,5,6,7,8] (strength of evidence V) (grade of recommendation 2B).

If molecular classification was not performed at the time of the biopsy, it must be performed for the final report (at least a basic variant of ProMisE) (grade of recommendation 2B).

LVSI invasion is a very important predictive factor indicating individual risk of recurrence and a decisive factor in the choice of adjuvant therapy [31] (strength of evidence IIC).

The semiquantitative LVSI assessment system, which distinguishes focal and substantial * LVSI depending on the number of vessels involved, confirmed the high agreement of the results [24] (strength of evidence IIC).

* Substantial LVSI signifies the involvement of more than five lymphovascular spaces (LVSI) and does not include LVSI within the tumour and in the immediate vicinity of the tumour margin [32] (strength of evidence IA).

The presence of substantial LVSI [32,33,34] (strength of evidence IB, IIIA and IIIE) is both predictive and prognostic and, therefore, the final histopathological result in the case of LVSI invasion should indicate if it is focal or substantial (grade of recommendation 1).

A detailed description of all clinically necessary elements of the histopathological report is included in Appendix A.

### 3.4. Imaging Prior to Treatment Decision

The best method of assessing the local advancement of endometrial cancer (the depth of the myometrial invasion and infiltration of the cervical stroma—pT2) is magnetic resonance imaging (MRI) with contrast [35,36] (strength of evidence V and IIIA). Expert ultrasound has a diagnostic value comparable to MRI in the assessment of myometrial infiltration but is significantly worse in the assessment of the T2 feature [35] (strength of evidence V). Computed tomography (CT) is only useful in assessing the spread of cancer beyond the pelvis. Radiological assessment of the pelvis by CT is inferior to MRI and expert ultrasound [36] (strength of evidence IIID).

Therefore, before deciding on the sequence of endometrial cancer treatment, clinical and radiological staging should be performed based on gynaecological examination, pelvic MRI and CT of the abdomen and the chest (grade of recommendation 2B).

In justified cases, expert ultrasound can replace magnetic resonance imaging in the assessment of changes in the pelvis (grade of recommendation 2B).

FIGO staging for endometrial carcinoma is shown in Table 5 [37].

## 4. Treatment

### 4.1. Clinical and Radiological Figo Stage I/II (Operable)

#### 4.1.1. Surgical Treatment

##### Uterus

Standard surgery is a simple total hysterectomy with bilateral salpingo-oophorectomy (BSO) **; peritoneal fluid sampling for cytological examination is not recommended [37,38] (strength of evidence V and IIIA) (grade of recommendation 2A).

As minimally invasive surgery (total laparoscopic hysterectomy (TLH) and total robotic hysterectomy (TRH)) does not compromise the prognosis and has a significant advantage in perioperative and postoperative outcomes over open surgery [39,40] (strength of evidence IIIA and IIIA), it is recommended where possible (grade of recommendation 2A).

Modified radical or radical hysterectomy (i.e., hysterectomy with a vaginal margin and/or partial/total parametrial resection) increases the number of complications and does not improve results [38] (strength of evidence IIIA). Therefore, performing such procedures and claiming them for medical reimbursement are not recommended (grade of recommendation 2A).

Exceptions are as follows:(1)Sparing treatment in women who want to preserve fertility and who have met the criteria * and achieved complete clinicopathological remission after hormone therapy [41] (strength of evidence IIIA) (grade of recommendation 2A).(2)No oophorectomy in women < 45 years old who have met the criteria ** [42,43,44] (strength of evidence IIIA, IIIA and IIIA) (grade of recommendation 2A).

* Criteria: no myometrial invasion (MRI with contrast) and exclusion of metastatic disease (CT of the abdomen and chest); bioptate G1 TP53wt or every G when POLE [45] (strength of evidence IIIA); and no contraindications for hormonal treatment and/or pregnancy (including age under 45) [41,46,47] (strength of evidence V, IIIA and IIIE) (a detailed algorithm of how to proceed is included in Appendix A).

** Criteria: age < 45; FIGO I/II—necessary exclusion of FIGO IIIC by systematic lymphadenectomy or sentinel lymph node procedure in the group at high risk of metastases in radiologically negative lymph nodes (see indications for lymphadenectomy); and high risk of ovarian cancer excluded such as for BRCA1/2 mutation carriers or hereditary nonpolyposis colorectal cancer (*HNPCC*) (Lynch syndrome).

##### Lymph Nodes (Staging)

Lymphadenectomy has no prognostic significance in endometrial cancer [48,49] (strength of evidence IIA and IIA).

A meta-analysis based on the results of a systematic review of randomised trials showed that removal of lymph nodes in radiological FIGO I patients, regardless of risk factors (histological type and grade), does not affect overall survival and the time-to-recurrence; it is, however, associated with a significant number of complications [50] (strength of evidence IA).

Therefore, lymphadenectomy should be considered in patients with substantial* (high-intermediate and high) risk of metastases in radiologically negative lymph nodes (grade of recommendation 1).

The goal of lymphadenectomy is to rule out the highly probable FIGO IIIC in this group of patients. The potential up-stage change is of prognostic significance [37] (strength of evidence V) and influences the choice of adjuvant treatment (see FIGO I/II adjuvant treatment vs. FIGO III adjuvant treatment for details).

Because the new molecular classification better defines the risk groups of metastases to the lymphatic system [20,21,22] (strength of evidence IIIE, IIIA and IIA) and there is a significantly greater risk of committing an error in making decisions about lymphadenectomy based on existing criteria (histopathological type, grading and MI) [23,24,25,26,27,28] (strength of evidence IIIC–IIID), before deciding to perform a lymphadenectomy in patients with radiological FIGO I/II, it is recommended to determine the risk of metastasis based on molecular criteria (grade of recommendation 2A), or if this is not yet possible, conditionally based on existing pathological and radiological features (grade of recommendation 3).

* Significant risk factors for metastases to the lymphatic system (indications for lymphadenectomy) are as follows:Low-grade endometroid carcinoma with myometrial invasion > 50% excluding POLEmut (high-intermediate risk);Non-endometroid carcinoma (high risk);TP53abn, every histological type (high risk);High-grade endometroid carcinoma excluding POLEmut (high risk) [20,21,22] (strength of evidence IIIE, IIIA and IIA) [51,52,53] (strength of evidence IIIA, IIID and IIIE) (grade of recommendation 2A);Lack of randomised trials comparing systematic lymphadenectomy with the sentinel lymph node procedure.

Furthermore, in the group of patients at high risk of metastasis, lymphadenectomy should include the removal of radiologically negative pelvic and para-aortic lymph nodes up to the left renal vein because it increases overall survival [53] (strength of evidence IIIE) (grade of recommendation 2A).

The sentinel lymph node procedure can be considered to assess the status of the nodes (pN) [54,55] (strength of evidence IIIB and IIIB) (grade of recommendation 2A).

Note: for the sentinel node procedure, pathomorphological ultrastaging of the excised lymph nodes is obligatory [55] (strength of evidence IIIB) (grade of recommendation 2A).

##### Greater Omentum

Infracolic omentectomy should be performed only in patients with serous or undifferentiated endometrial carcinoma due to the increased risk of omental metastasis that occurs in these histopathological types [56] (strength of evidence IIIA) (grade of recommendation 2A).

#### 4.1.2. Adjuvant Treatment (Post-Surgery Patients with No Residual Disease R0)

Radiotherapy is the method of choice [57] (strength of evidence IIA). Chemotherapy is not recommended as the adjuvant treatment of FIGO I/II endometrial cancer (this also applies to the non-endometroid types, including serous carcinoma) [58] (strength of evidence IIIA) (grade of recommendation 1).

The decision to proceed with adjuvant radiotherapy should be made after determining the individual risk of recurrence based on molecular type, grading, LVSI and MI [59] (strength of evidence 5).

For low-risk patients, observation is recommended [60] (strength of evidence IIA) (grade of recommendation 1).

For patients at intermediate and high-intermediate risk, brachytherapy (BT) is recommended [61,62] (strength of evidence IIA and IIA) (grade of recommendation 1).

For patients at high risk, brachytherapy (BT) and external beam radiation therapy (EBRT) are recommended (strength of evidence V) (grade of recommendation 1).

### 4.2. Clinicoradiological FIGO Stage I/II (Inoperable—Not Suitable for Surgery Due to Health Issues)

There is a lack of randomised studies comparing different methods of radiotherapy (EBRT/BT) in the radical treatment of inoperable FIGO stage I/II endometrial cancer.

Retrospective cohort studies have shown that the use of BT HDR in the radical treatment of inoperable endometrial cancer improves overall survival (OS) and progression-free survival (PFS) time [63] (strength of evidence IIIE). The prognosis of patients undergoing independent EBRT (without BT) may be worse [64] (strength of evidence IIIE).

Other studies suggest that adding EBRT to BT does not improve results [65,66] (strength of evidence IIIE, IIIE) and significantly increases the toxicity of treatment [66] (strength of evidence IIIE). However, these are retrospective studies assessing the treatment of patients with various doses at a time when modern irradiation techniques were not used.

There is a need to conduct prospective randomised trials comparing standalone BT, standalone EBRT and combo therapies: BT and EBRT.

Until an unequivocal answer is obtained as to which option is the most favourable, the treatment of choice safest for patients is always the BT HDR variant, supplemented in the group at high risk of metastases to lymph nodes with EBRT (using modern irradiation techniques).

Therefore, in patients who are not eligible for surgery (poor general condition or lack of consent to surgery) with significant risk factors for metastasis to the lymph nodes, the following are recommended:Low-grade endometroid carcinoma with myometrial invasion > 50% excluding POLEmut (high-intermediate risk).Non-endometroid carcinoma (high risk).TP53abn, every histological type (high risk).High-grade endometroid carcinoma excluding POLEmut (high risk) [20,21,22] (strength of evidence IIIE, IIIA and IIA) [51,52,53] (strength of evidence IIIA, IIID and IIIE). The treatment of choice is BT (uterus) combined with EBRT (uterus and lymph nodes).

In other cases, only BT should be used (expert opinion) (strength of evidence V) (grade of recommendation 2B).

It is recommended that the intensity-modulated radiation therapy (IMRT)/VMAT technique or conformal radiotherapy be used in the area of the reproductive organs and lymph nodes. HDR brachytherapy should be planned based on CT or MRI performed after the insertion of the applicator (expert opinion) (strength of evidence V) (grade of recommendation 2B).

### 4.3. Clinicoradiological FIGO Stage IIIA/B/C/IVA (Operable Locally Advanced Endometrial Carcinoma)

#### 4.3.1. Surgical Treatment

##### FIGO IIIA—Adnexal Invasion

Simple TLH/TRH/TAH with BSO is recommended; peritoneal fluid sampling for cytological examination and systematic lymphadenectomy is not recommended [67] (strength of evidence IIID) (grade of recommendation 2A).

##### FIGO IIIB, IVA—Infiltration of Structures Adjacent to the Uterus: Parametrium, Bowel and Bladder

Cytoreductive surgery in radiologically advanced endometrial cancer (FIGO IIIB and IVA) is allowed only when the patient’s general condition is good (the patient qualifies for major surgery) and when the operator can perform complete cytoreduction (lack of residual disease R0—microscopically negative margins of resection) [67] (strength of evidence IIID) (grade of recommendation 2A).

If the disease extends beyond the uterus but does not exceed the pelvic boundaries (rectal/sigmoid infiltration and/or *parametrium), the recommended method of surgical treatment is en-block cancer resection with reconstruction of the gastrointestinal tract or * radical hysterectomy (expert opinion) (strength of evidence V) (grade of recommendation 2B).

Systematic lymphadenectomy, in these cases, is not recommended [49,50] (strength of evidence IIA and IA) (grade of recommendation 3).

##### FIGO IIIC—Pelvic and/or Para-Aortic Lymph Nodes Radiologically Suspected of Metastasis

Simple TAH with BSO is recommended without peritoneal fluid sampling for cytological examination and with a biopsy of the suspected lymph nodes or, where possible, a selective lymphadenectomy (authors’ opinion) (strength of evidence V) (grade of recommendation 2B).

#### 4.3.2. FIGO III Adjuvant Treatment (Post-Surgery with No Residual Disease R0)

CAUTION: This applies to preoperative apparent FIGO I/II cases in which FIGO IIIA or IIIC was diagnosed postoperatively as a result of surgical staging and successfully resected cases of operable FIGO IIIB.

Radiochemotherapy is a method of choice (operable FIGO IIIA, B and C endometrial carcinoma, every histopathological type) [68,69] (strength of evidence IIIA and IIA) (grade of recommendation 1).

The recommended sequence of radiochemotherapy is two cycles of cisplatin with EBRT followed by four cycles of carboplatin with paclitaxel [69] (strength of evidence IIA) (grade of recommendation 1).

It is allowed to reverse the sequence of the treatment regimen: four cycles of carboplatin with paclitaxel and then two cycles of cisplatin with EBRT (expert opinion—strength of evidence V) (grade of recommendation 2B). In POLE and MMRd endometrial cancer subtypes, EBRT alone is recommended because the addition of chemotherapy (CHT) before and during irradiation is of no benefit in this group of patients [69] (strength of evidence IIA) (grade of recommendation 1).

### 4.4. Clinical and Radiological FIGO Stage IIIA/B/C/IVA (Inoperable or Unresectable Locally Advanced Cancer)

In patients with locally advanced cancer without distant metastases (note: M1—FIGO IVB: the presence of metastases outside the pelvis or in the nonregional lymph node) who are not eligible for surgery (poor general condition, no consent to surgery and/or complete cancer resection is impossible), the method of choice is EBRT including uterus and lesions in the pelvis depending on their location (parametrium, adnexae and pelvic lymph nodes) and/or metastatic para-aortic lymph nodes combined with BT (uterus) (expert opinion) (strength of evidence V) (grade of recommendation 2B).

It is recommended that the IMRT/VMAT technique or conformal radiotherapy be used in the area of reproductive organs and pelvic lymph nodes. HDR BT should be planned based on CT or MRI performed after the insertion of the applicator. In cases where radical treatment is not feasible, the method of choice is systemic treatment with or without palliative radiotherapy (expert opinion) (strength of evidence V) (grade of recommendation 2B).

### 4.5. Locally Advanced Cancer with Residual Disease after Surgery (Incompletely Resected FIGO III-IVA), Metastatic Disease (Note: M1—FIGO IVB: Presence of Metastases Outside the Pelvis or in the Nonregional Lymph Node) and Recurrence

In patients with incomplete resection (FIGO III—IVA, R1/R2) or disseminated cancer, i.e., cases with unresectable metastases to the lungs or liver with a multifocal spread in the abdominal cavity (FIGO IVB) or with unresectable recurrence, the method of choice is systemic treatment (+/− radiotherapy) [70,71,72,73,74,75,76] (strength of evidence IIIA, IIIA, IIA, IID, IIA, IID and IID) (grade of recommendation 1).

The type of therapy should be selected individually considering the histological type, receptor status and/or molecular profile.

#### 4.5.1. Variants of Systemic Treatment

##### Hormonotherapy

For patients with low-grade endometroid carcinoma, it is recommended to determine the expression of E/P receptors (receptor status) because hormone therapy is a preferred option in this group (the percentage of objective responses (ORR) is 21.6%: ER + 26.5% and PR + 35.5%) (does not apply to cases with rapid progression of the disease) [71] (strength of evidence IIIA) (grade of recommendation 2A).

Progestogens: megestrol acetate at a dose of 160 mg/ day or medroxyprogesterone at a dose of 200 mg/day are recommended. Progestogens can be alternated with tamoxifen [70,71] (strength of evidence IIIA and IIIA) (grade of recommendation 2A).

The percentage of objective response rates (ORR) after the use of aromatase inhibitors is low (7%), but due to the high (44%) clinical benefit (stabilisation and response) [77] (strength of evidence II D), their use may be considered in selected cases (grade of recommendation 2B).

##### First-Line Chemotherapy

For patients with high-grade endometroid carcinoma and non-endometroid carcinoma (serous, clear cell and carcinosarcoma), a carboplatin and paclitaxel regimen is recommended [72] (strength of evidence IIA) (grade of recommendation 1).

##### Trastuzumab in Serous Carcinoma

In a randomised phase II trial [78] (strength of evidence IIA), patients with advanced/metastatic (FIGO III-IV) or recurrent HER2-positive serous endometrial cancer received carboplatin and paclitaxel with or without trastuzumab and continued treatment with trastuzumab until disease progression or unacceptable toxicity.

In the arm of patients receiving trastuzumab, a significantly longer median PFS was achieved, reaching 12.6 vs. 8.0 months (HR 0.44, 95% CI 0.26 to 0.76; *p* = 0.005) [78] (strength of evidence IIA), as well as an improvement in the median time of overall survival by 5.2 months (HR 0.58, 95% CI 0.34 to 0.99; *p* = 0.046) [79] (strength of evidence IIA).

A particular benefit of trastuzumab was observed in the first line of treatment with PFS of 17.9 vs. 9.3 months (HR 0.40, 90% CI 0.20 to 0.80; *p* = 0.013) [78] (strength of evidence IIA) and with a median OS not reached in the trastuzumab arm compared to 24.4 months in the control group (HR 0.49, 90% CI 0.25 to 0.97; *p* = 0.041) [79] (strength of evidence IIA). Adding trastuzumab to chemotherapy did not increase the toxicity of the treatment.

Therefore, in patients with advanced/metastatic or recurrent serous cancer, it is recommended to determine the status of the HER2 receptor because the addition of trastuzumab to chemotherapy (carboplatin and paclitaxel) is the most favourable (recommended) therapeutic option in the group of HER2-positive patients (grade of recommendation 1).

The standards for determining the status of the HER2 receptor are included in Appendix A.

##### Second-Line Chemotherapy

Based on the results of a randomised phase III trial [74] (strength of evidence IIA) indicating a significant advantage of immunotherapy (pembrolizumab plus lenvatinib) over chemotherapy (paclitaxel or doxorubicin) (reduction in the risk of recurrence by 46% (HR of 0.54) and the risk of death by 38 % (HR of 0.62)), it is recommended to use chemotherapy only in recurrent sarcoma (grade of recommendation 1).

In other histological types, the use of chemotherapy (paclitaxel or doxorubicin) may only take place in clinically justified situations or when there are limitations in the availability of immunotherapy (grade of recommendation 3).

When choosing chemotherapy, the patient should be informed of a significantly worse prognosis (expert opinion) (grade of recommendation 2B).

##### Immunotherapy

In the entire population of patients with recurrent or advanced endometrial cancer, regardless of MMR/MSI status, the use of a combination of a PD-1 inhibitor (pembrolizumab) with a multiply tyrosine kinase inhibitor against VEGFR1, VEGFR2 and VEGFR3 (lenvatinib) was superior to chemotherapy (reduction in the risk of recurrence by 46% (HR of 0.54) and the risk of death by 38% (HR of 0.62)). The objective response rate (ORR) was 32%. The rate of serious adverse events was 89%, and 33% of patients discontinued therapy [74] (strength of evidence IIA).

In a single-arm study [73] (strength of evidence IID) and in a phase II study [75] (strength of evidence IIC), monotherapy with PD-1 inhibitors (dostarlimab and pembrolizumab) showed high effectiveness in the treatment of patients with recurrent, advanced or metastatic endometrial cancer with mismatch repair deficiency (MMRd)/high microsatellite instability (MSI-H) that have progressed during or after platinum-based therapy.

The objective response rate (ORR) was 44% for dostarlimab [73] (strength of evidence IID) and 57% (D arm) or 48% (D and K arm) for pembrolizumab [75,76] (strength of evidence IIC and IIC).

The rate of treatment-related serious adverse events was similar for both PD-1 inhibitors (dostarlimab—11.5%; pembrolizumab—12%) [73] (strength of evidence IID) [75] (strength of evidence IIC).

Therefore, in patients with incomplete resection of locally advanced cancer (FIGO III—IVA, R2) or disseminated cancer (FIGO IVB) or with unresectable recurrence who progressed after platinum-based chemotherapy (at least one cycle), the treatment of choice is immunotherapy [73,74,75] (strength of evidence IIA, IIC and IID) (grade of recommendation 1).

MMRd)/MSI-H cancers may be treated with PD-1 inhibitors (dostarlimab or pebrolizumab) or with a combination of PD-1 inhibitor (pembrolizumab) and lenvatinib (grade of recommendation 1). Due to the high percentage of treatment discontinuation observed in the combo therapy of pembrolizumab and lenvatinib, monotherapy with PD-1 inhibitors is the preferred option in this group of patients (expert opinion—strength of evidence V) (grade of recommendation 2B).

In the group of patients with a mismatch repair proficiency (MMRp), the treatment of choice is a combination of a PD-1 inhibitor (pembrolizumab) with lenvatinib (strength of evidence IIA) (grade of recommendation 1).

## 5. Follow-Up

There is no evidence that any post-treatment surveillance regimen for endometrial cancer improves patient survival time.

The randomised TOTEM study assessed the role of intensive (INT) vs. minimalist (MIN) surveillance after the treatment of patients with endometroid carcinoma of the endometrium [80] (strength of evidence IIA). Patients were stratified into low- (FIGO IA, low-grade) and high-risk (≥FIGO IA, high-grade) groups for recurrence. There were no statistically significant differences in survival and time-to-recurrence in patients under minimal and intensive surveillance in both the low- and high-risk groups.

The TOTEM study looked only at endometroid cancer and did not consider the molecular criteria for assessing the risk of disease recurrence, including TP53 status (Table 6).

In the opinion of experts, patients with non-endometroid carcinoma and those with endometroid carcinoma in whom an abnormal/mutated P53 gene was detected should be subject to intensive surveillance (expert opinion) (strength of evidence V) (grade of recommendation 2B).

Extrapolating the results of the TOTEM study [80] and expert suggestions, the recommended follow-up schedule adjusted to the current risk groups for recurrence [59] is described in Table 7 (expert opinion) (strength of evidence V) (grade of recommendation 2B).

## 6. Hereditary Endometrial Carcinoma

Hereditary nonpolyposis colorectal cancer (*HNPCC)* (Lynch syndrome).

A total of 30–40% of endometrial cancer patients have MMRd/MSI-H) [12] (strength of evidence V).

In this group, every 8th woman (5% of endometrial cancer cases) has a germinal mutation in the *MSH2, MLH1, MSH6 and PMS2* genes coding for DNA mismatch repair (MMR) proteins [81] (strength of evidence V) and *EPCAM* deletions (*EPCAM-MSH2*) and thus has Lynch syndrome [82] (strength of evidence IVA).

*MSH2, MLH1, MSH6* and *PMS2* mutations are detected, respectively, in 41, 37, 13 and 9% of Lynch syndrome cases [83] (strength of evidence V). *EPCAM* deletions (*EPCAM-MSH2*) are detected in 3% of Lynch syndrome cases [82,84] (strength of evidence IVA and IVA).

The lifetime risk of developing colorectal cancer in the female population is 4.7% [85] (strength of evidence V). In women with Lynch syndrome, the risk is on average 10 times higher and, depending on the mutation, it ranges between 10% and 45% [84,85] (strength of evidence IVA and V). The highest risk is for mutations in *MLH1* (45%) and *MSH2* (33%) and the lowest for mutations in *MSH6* (26%) and *PMS2* (10–12%) [86,87] (strength of evidence IIIB and IIIB).

The lifetime risk of developing endometrial cancer in the general population is 2.9% [85] (strength of evidence V). In women with Lynch syndrome, it is on average 10 times higher and, depending on the mutation, it ranges between 21% and 51% [85] (strength of evidence V). The highest risk is for carriers of the mutations in *MSH2* (51%) and *MSH6* (49%) and the lowest for the mutations in *MLH1* (34%) and *PMS2* (12–24%) [85,86,87,88] (strength of evidence V, IIIB, IIIB and IIIB).

Hereditary endometrial cancer is diagnosed in the fourth and fifth decades of life. Carriers of the *MSH2* mutation are affected the earliest (mean age 47) and *MSH6* mutation carriers the latest (mean age 53) [85] (strength of evidence V).

Lynch syndrome also increases the risk of developing ovarian cancer (3%–20% compared to 1.3% risk in the general population) [89] (strength of evidence IIIB).

Genetic testing for Lynch syndrome should be performed in all cases of endometrial cancer with MMRd/MSI-H regardless of age, histological type [90] (strength of evidence IIID) (grade of recommendation 2A), and Amsterdam Criteria (expert opinion) (strength of evidence V) (grade of recommendation 2B).

The Amsterdam Criteria II (screening for Lynch syndrome in the general population) are as follows:At least three relatives with a Lynch-associated cancer (colorectal, endometrial, small intestine, ureter and renal pelvis) verified pathologically;One is a first-degree relative of the other two;At least two successive generations affected;At least one relative is diagnosed before the age of 50;Familial adenomatous polyposis has been ruled out;Tumours should be verified by pathologic examination [91] (strength of evidence V).

Other rare hereditary syndromes predisposing to endometrial cancer are described below.

### 6.1. Cowden Syndrome

PTEN germline mutations (<1%) increase the risk of developing endometrial cancer by 25% [72] (strength of evidence V). It does not affect the risk of ovarian cancer [92] (strength of evidence IIIA).

### 6.2. Hereditary Breast Cancer Side Specific (HBss), Breast and Ovarian Cancer (HBOC) and Ovarian Cancer (HOC) Syndrome

Germline mutations in BRCA1/2 are detected in 1% endometrial cancer. Carriers of the BRCA1 germline mutation often show a loss of the second copy of the gene, which causes a total biallelic inactivation of the BRCA1 protein, which is manifested by a deficiency in DNA double-strand break repair (HRD) and susceptibility to treatment with platinum compounds and PARP inhibitors. In contrast, biallelic inactivation of *BRCA2* is rare [93] (strength of evidence IVA).

Among carriers of germline mutations in *BRCA1/2* genes, the absolute risk of developing endometrial cancer up to 75 years of age is 3% (similar to life risk in the general population—2.9% [76] (strength of evidence V)), and for the serous type, 1.1% [93] (strength of evidence IIIA).

Confirmation of mutations in *MMR* and *EPCAM* deletions (*EPCAM-MSH2*) (Lynch syndrome) in *BRCA1/2* (HBss/HBOC/HOC) and in PTEN (Cowden’s syndrome) should result in genetic testing among relatives to identify carriers of the germline mutation.

In the case of Lynch syndrome, each family member with a confirmed mutation (regardless of gender) should undergo colonoscopy screening according to the Polish Society of Oncological Surgery (PSOS) recommendations (expert opinion) (strength of evidence V) (grade of recommendation 2B).

Women (Lynch syndrome and Cowden syndrome) should be encouraged to have children early and be offered prophylactic hysterectomy after the completion of reproductive plans [86,87,92] (strength of evidence IIIB, IIIB and IIIA) (grade of recommendation 2A).

Before a prophylactic hysterectomy, the presence of endometrial, ovarian and colorectal cancer should be excluded. For this purpose, endometrial biopsy, transvaginal ultrasound of ovaries with the determination of serum CA125 and colonoscopy are recommended (expert opinion) (strength of evidence V) (grade of recommendation 2B).

Prophylactic oophorectomy and salpingectomy are recommended in Lynch syndrome [86,87] (strength of evidence IIIB and IIIB) and in the case of detection of the germinal mutation *BRCA1/2* (HOC/HBOC/HOC) [94] (strength of evidence IIIA) (grade of recommendation 2A).

In the case of *PTEN* (Cowden syndrome) germinal mutation, a prophylactic oophorectomy and salpingectomy are not recommended [92] (strength of evidence IIIA) (grade of recommendation 2A).

In the case of the *BRCA1/2* germline mutation (HOC/HBOC/HOC), prophylactic hysterectomy is not recommended [94] (strength of evidence IIIA) (grade of recommendation 2A).

## Figures and Tables

**Table 1 jcm-12-01480-t001:** Grading criteria according to The Agency for Health Technology Assessment and Tariff System (AOTMiT) guidelines.

Study Type	Grade	Subtype Description
RTC systematic review	IA	Meta-analysis based on RTC systematic review results
	IB	RCT systematic review without meta-analysis
Experimental study	IIA	Well- conducted randomised controlled trial, including pragmatic randomised controlled trial
	IIB	Well-conducted clinical controlled trial with pseudorandomisation
	IIC	Well-conducted clinical controlled trial without randomisation
	IID	One-arm study
Observational study with control group	IIIA	Systematic review of observational studies
	IIIB	Well-conducted prospective cohort studies with simultaneous control group
	IIIC	Well-conducted prospective cohort studies with historic control group
	IIID	Well-conducted retrospective cohort studies with simultaneous control group
	IIIE	Well-conducted case-control study (retrospective)
Descriptive study	IVA	Case series—pretest/posttest *
	IVB	Case series—posttest **
	IVC	Other study of a group of patients
	IVD	Case report
Expert opinion	V	Expert opinions based on clinical experience and reports from expert panels

* Pretest/posttest—a study where measurements are taken both before and after the assessed intervention. ** Posttest—a study where measurements are taken only after the intervention.

**Table 2 jcm-12-01480-t002:** Polish Society of Gynaecological Oncology (PSGO) recommendation classification system.

Grade of Recommendation	Grading Criteria (Strength of Evidence)
Grade 1	Strength of evidence I or II (unanimity of experts) *
Grade 2A	Strength of evidence III (unanimity of experts) *
Grade 2B	Strength of evidence IV or V (unanimity of experts) *or strength of evidence III (no unanimity of experts) *
Grade 3	Every strength of evidence, when PSGO development group believes that the procedure can be used under certain conditions but is not appropriate (unanimity) *

* Unanimity: ≥ 85% of development group members agree.

**Table 3 jcm-12-01480-t003:** Histopathological classification of the carcinoma of the endometrium.

Original Name	ICD-O Code	Frequency (Percentage)	Definition	Prognosis
Endometrioid carcinoma (EEC), NOS *POLE*-ultramutated EEC, mismatch repair-deficient EEC, P53-mutant EEC, NSMP * EEC	8380/3	About 85%	Adenocarcinoma with a glandular architecture imitating the structure of the endometrium and (depending on grade: G1–G3) a solid component. Develops on the basis of atypical endometrial hyperplasia, most often in women with high body mass index (BMI) and hyperestrogenism. Positive for ER/PR. Its etiopathogenesis is related to the activation of signalling pathways: PI3K-PTEN-AKT-mTOR, RAS-MEK-ERK and WNT-β-catenin, microsatellite instability or gene mutation: *POLE* or *ARID1A*. There are four molecular subtypes in the group of endometroid carcinomas (Table 4).	Most EEC cancers are low-grade and have good prognosis. Percentage of G3, with p53 overexpression and negative for ER, does not exceed 10–19% of all cases.
Serous carcinoma, NOS	8441/3	3–10%	Cancer consisting mostly of cells with high degree of atypia, forming papillae and microtubule systems and/or glandular systems. It develops on the basis of the endometrial polyp or atrophic endometrium in postmenopausal women. In most cases, mutation of *P53* gene can be observed; in 30% of cases, *ERBB2* (*HER2*) gene amplification. Most often negative for ER and PR; however, that can be variable.	Aggressive behaviour, responsible for 40% of endometrial cancer deaths
Clear cell adenocarcinoma NOS	8310/3	<5%–10%	Cancers showing significant diversity, both in terms of cell morphology (with clear or eosinophilic cytoplasm and various shapes, including the so-called hobnail cells) and the structures formed (papillary systems with hyalinised core, tubular, cystic and solid). Cell atypia is variable, but at least focally, there is high-grade atypia. All clear cell carcinomas are grade G3. Usually negative for ER and PR.	Aggressive behaviour
Undifferentiated and dedifferentiated carcinoma	8020/3		Undifferentiated carcinoma is a malignant epithelial tumour without specific morphologic evidence of epithelial differentiation. Cancer cells form a solid-pattern tumour. Negative for ER/PR. Common mutations of many genes, including *P53* and *POLE,* and in half of the cases, features of microsatellite instability. In the case of dedifferentiated carcinoma, the undifferentiated component is accompanied by endometrioid carcinoma of various grades (G1-G3) or serous carcinoma.	Aggressive behaviour
Mixed cell adenocarcinoma	8323/3	About 10%	Cancer composed of two or more histological subtypes of endometrial carcinoma, at least one of which is serous or clear cell carcinoma. Each of the components must account for at least 5% of the tumour tissue.	Aggressive behaviour
Mesonephric adenocarcinoma	9110/3	Extremely rare	Adenocarcinoma originating from the remains of the mesonephrium, i.e., Wolff’s ducts. Found rarely in the uterine corpus, more often in the cervix. Negative for ER/PR; they express calretinin and antigens: CD10 and GATA3.	The number of cases described so far is too small to assess tumour biology
Squamous cell carcinoma NOS	8070/3	<0.5%	Cancer formed exclusively from atypical squamous cells.	Prognosis depends on the grade; in most cases, poor
Mucinous carcinoma, intestinal type	8144/3	Extremely rare	A rare histological form of endometrial cancer; a neoplasm with a high degree of differentiation consisting of cells capable of producing mucilage stained with mucicarmine and expressing the CEA antigen.	Prognosis for patients diagnosed with FIGO stage I-II is relatively good
Mesonephric-like adenocarcinoma	9111/3	Extremely rare	Extremely rare type of endometrial cancer; until 2021, only 115 cases were reported. Histologically and immunophenotypically imitates mesonephric adenocarcinoma. Negative for ER/PR.	Aggressive behaviour, at least half of the cases are diagnosed in advanced stage (FIGO ≥ II)
Carcinosarcoma NOS	8980/3	About 5%	Biphasic neoplasm with a high-grade carcinoma component (most often EEC or serous carcinoma) and a sarcoma component. The sarcoma component most often takes the form of a high-grade undifferentiated sarcoma but may also be heterologous—rhabdomyosarcoma, chondrosarcoma and osteosarcoma. Molecular studies have shown that both tumour components present the same genetic aberrations. The mesenchymal component is assumed to be derived from the epithelial component and formed through EMT (epithelial-mesenchymal transition) in the course of metaplastic conversion. Most (90%) cases show mutations in the P53 gene. Tumour metastases in about 90% of cases are cancerous.	Aggressive behaviour

* No specific molecular profile.

**Table 4 jcm-12-01480-t004:** Molecular classification of the carcinoma of the endometrium (refers to endometrioid carcinoma, EEC; serous carcinoma; and carcinosarcoma).

	*POLE*-Ultramutated EC (*POLE* EDM)	MMR-Deficient (MSI) EC (MMRd/MSI-H)	*P53*-Mutated EC (CNhigh)	NSMP EC (CNlow)
Frequency (percentage)	6–9%	20–30%	12–18%	40–50%
Number of mutations/ Mb	>100	10–100	<10	<10
Copy number variation (SCNA)	Very low	Low	High (substantial)	Low
Other molecular characteristics	MSS	MSI	MSS	MSS and loss of *CTNNB1* in 30–40% of cases
5 most common mutations (percentage of tumours with mutation)	*POLE* (100%), *DMD* (100%), *CSMD* (100%), *FAT4* (100%), *PTEN* (94%)	*PTEN* (88%), *PIK3CA* (54%), *PIK3R1* (42%), *RPL22* (37%), *ARID1A* (37%)	*P53* (92%), *PIK3CA* (47%), *FBXW7* (22%), *PPP2R1A* (22%), *PTEN* (10%)	*PTEN* (77%), *PIK3CA* (53%), *CTNNB1* (52%), *ARID1A* (42%), *PIK3R1* (33%)
Histological subtype	Endometrioid carcinoma(G1–G3)	Endometrioid carcinoma (G1–G3)	Serous carcinoma Endometrioid carcinoma (G3) Carcinosarcoma	Endometrioid carcinoma(G1–G2)
Selected histological features	Often high-grade and histologically ambiguous with presence of giant cells, pushing tumour border configuration of growth, substantial TILs	Often high-grade, substantial TILs, mucosal differentiation features, MELF invasion, substantial LVSI inflammatory infiltration around the neoplastic tumour	May be high-grade with diffuse nuclear and cytoplasmic atypia, coexistence of solid and glandular formation, destructive infiltration of uterine wall, LVSI	Most often low-grade, features of squamous cell differentiation present, absence of TILs, positive for ER/PR, p53 wild type expression
Diagnostic test	NGS; SNaPshot, qPCR; hotspot mutations: p.Pro286Arg, pVal411Leu, pSer297Phe, pAla456Pro, pSer459Phe, p.Phe367Ser, p.Leu424Ile, p.Met295Arg, p.Pro436Arg, p.Met444Lys, p.Asp368Tyr; tumour mutation burden (TMB)	IHC evaluating the presence of proteins MLH1, MSH2, MSH6 and i PMS2; MSI assay; NGS; tumour mutation burden (TMB)	Presence of p53 protein (mutant type) SCNA test	MMR-proficient, p53 wild type expression
Accompanying clinical features	Younger age, low body mass index (BMI)	Can be associated with Lynch syndrome, high body mass index (BMI)	Advanced stage at the time of diagnosis, low body mass index (BMI)	High body mass index (BMI)
Prognosis in the early stage of the disease (FIGO I–II) (5-year survival without recurrence)	Excellent (100%)	Fair (about 80%)	Poor (about 50%)	Variable (from fair to excellent) (about 65%)

**Table 5 jcm-12-01480-t005:** FIGO staging for endometrial carcinoma.

FIGO 2009	
I	Cancer limited to the body of the uterus (T1)
IA	No or less than half (≤50%) myometrial invasion
IB	Invasion equal to or more than half (≥50%) of the myometrium
II	Cancer invades cervical stroma but does not extend beyond uterus. Endocervical glandular involvement only is stage I
III	Local and/or regional spread of the cancer: invasion of structures next to the uterus (T3) or regional lymph node involvement (N1)
IIIA	Cancer invades the serosa of the body of the uterus and/or adnexa (T3A)
IIIB	Vaginal and/or parametrial involvement and/or bowel without mucosa and/or bladder without mucosa (T3B)
IIIC	Regional lymph nodes metastases (N1)
IIIC1	Positive pelvic nodes
IIIC2	Positive para-aortic lymph nodes with or without positive pelvic lymph nodes
IV	Cancer invades bladder and/or bowel mucosa (T4) and/or distant metastases (M1) and/or nonregional lymph node involvement (N2)
IVA	Cancer invasion bladder and/or bowel mucosa (T4)
IVB	Distant metastases/beyond the pelvis (M1) and/or to nonregional lymph nodes (N2)

**Table 6 jcm-12-01480-t006:** Risk groups for endometrial carcinoma (applies only to postsurgery patients with no residual disease (R0)).

Risk Group	Characteristics
Low	FIGO IA endometroid low-grade (MMRd ^a^ and NSMP) with negative or focal LVSI. FIGO I/II *POLE*mut; FIGO III *POLE*mut ^b^
Intermediate	FIGO IA endometroid high-grade (MMRd and NSMP) with negative or focal LVSI. FIGO IA TP53abn and/or non-endometroid (serous, clear cell, undifferentiated carcinoma, carcinosarcoma and mixed) with no myometrial invasion and with negative or focal LVSI. FIGO IB endometroid low-grade (MMRd and NSMP) with negative or focal LVSI.
High intermediate	FIGO I endometroid type (MMRd and NSMP) (regardless of grade and depth of invasion) with substantial LVSI. FIGO IB endometroid type (MMRd and NSMP) high-grade regardless of LVSI status. FIGO II endometroid type (MMRd and NSMP).
High	Every FIGO stage and every histological type invading myometrium with TP53 abn. Every FIGO stage with serous carcinoma or undifferentiated carcinoma including carcinosarcoma with myometrial invasion. FIGO III-IVA endometroid type (MMRd and NSMP) with no residual disease (R0).

^a^ MMRd and MSI-H-: both terms identify the same endometrial cancer patient population (identification by IHC for MMRd and sequencing for MSI-H). ^b^ FIGO III *POLE*mut can be classified as a low-risk group; however, there is insufficient data that would allow the avoidance of adjuvant treatment in this group.

**Table 7 jcm-12-01480-t007:** Recommended post-treatment follow-up schedule for patients with endometrial cancer.

Risk Group	Follow-Up
Low: FIGO IA endometroid low-grade (MMRd ^a^ and NSMP with negative or focal LVSI. FIGO I/II *POLE*mut; FIGO III *POLE*mut ^b^	- For first 5 years after completing standard treatment, clinical visit* every 6 months. - (MIN)
Intermediate: FIGO IA endometroid high-grade (MMRd and NSMP) with negative or focal LVSI. FIGO IA TP53abn and/or non-endometroid (serous, clear cell, undifferentiated carcinoma, carcinosarcoma and mixed) with no myometrial invasion and with negative or focal LVSI. FIGO IB endometroid low-grade (MMRd and NSMP) with negative or focal LVSI.	- For first 2 years after completing standard treatment: clinical visit * every 4 months; CT of chest, abdomen and pelvis every 12 months - From the third to the fifth year of surveillance, clinical visit * every 6 months. - (MIN)
High intermediate: FIGO I endometroid type (MMRd and NSMP) (regardless of grade and depth of invasion) with substantial LVSI. FIGO IB endometroid type (MMRd and NSMP) high-grade regardless of LVSI status. FIGO II endometroid type (MMRd and NSMP).	
High: Every FIGO stage and every histological type invading myometrium with TP53 abn. Every FIGO stage with serous carcinoma or undifferentiated carcinoma including carcinosarcoma with myometrial invasion. FIGO III-IVA endometroid type (MMRd and NSMP) with no residual disease (R0).	- For first 3 years after completing standard treatment, clinical visit *; Ca125; transvaginal and abdominal ultrasound every 4 months (except when coinciding with abdominal CT); cytology; and CT of the abdomen and pelvis every 12 months - In fourth and fifth year of surveillance, clinical visit *; Ca125; transvaginal and abdominal ultrasound every 6 months (except when coinciding with abdominal CT); cytology; and CT of chest, abdomen and pelvis every 12 months

* Clinical visit with gynaecological examination (bimanual and speculum). ^a^ MMRd and MSI-H-: both terms identify the same endometrial cancer patient population (identification by IHC for MMRd and sequencing for MSI-H). ^b^ FIGO III *POLE*mut can be classified as a low-risk group; however, there is insufficient data that would allow the avoidance of adjuvant treatment in this group.

## Data Availability

Not applicable.

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
