# Peer review of "The Polish Society of Gynecological Oncology Guidelines for the Diagnosis and Treatment of Endometrial Carcinoma (2023)"

_jcm, 2023, doi:10.3390/jcm12041480_

Round 1
Reviewer 1 Report
I would like to congratulate the authors for this excellent overview about the modern therapy of endometrial carcinoma.
Just 2 aspects could be better described oder added.
1 - Regarding Nodal Staging:
- the role of Sentinel node Biopsy is not mentioned. It must be included in the guidelines, even if not enough evidence is avaiable.
- Extension of nodal dissection (only PA or PA + pelvic) must be described.
2 - Page 11
4.2. CLINICO-RADIOLOGICAL FIGO STAGE I/II (inoperable)
Please clarify inoperable in Stage I/II
Thank you for the opportunity to review this manuscript.
Author Response
1 - Regarding Nodal Staging:
- the role of Sentinel node Biopsy is not mentioned. It must be included in the guidelines, even if not enough evidence is avaiable.
- Extension of nodal dissection (only PA or PA + pelvic) must be described
Answer: The extent of lymphadenectomy has been described in guidelines (based on available evidence). Full nodal staging is required only in high risk patients and must include PA + pelvic.
In other groups of patients, sentinel lymph node procedure can be considered which is a form o selective pelvic lymphadenectomy.
(…) In the group of patients at high risk of metastasis, lymphadenectomy should include the removal of radiologically negative pelvic and para-aortic lymph nodes up to the left renal vein, because it increases overall survival [52] [Strength of evidence IIIE] (grade of recommendation 2A). There are no randomized trials comparing systematic lymphadenectomy with the sentinel lymph node procedure. Sentinel lymph node procedure can be considered to assess the status of the nodes (pN) [53, 54] [Strength of evidence IIIB, IIIB] (grade of recommendation 2A).
Note: for the sentinel node procedure, pathomorphological ultrastaging of the excised lymph nodes is obligatory [54] [Strength of evidence IIIB] (grade of recommendation 2A). (…)
2 - Page 11
4.2. CLINICO-RADIOLOGICAL FIGO STAGE I/II (inoperable)
Please clarify inoperable in Stage I/II
Answer
It means that patient is not suitable for surgery due to health issues
This explanation was added to main text of guidelines
Reviewer 2 Report
Dear Authors,
It was a pleasure to review your manuscript. This publication is much awaited and needed in the field. It includes up-to-date knowledge shared in comprehensive way. It is one of a few manuscripts that cover the new pathological grading system for endometrial cancer. I'm sure it will be big help for gyn-oncologists both in your country and around the globe.
- What is the main question addressed by the research? Is it relevant and interesting?
- The main question is how we should diagnose and treat endometrial cancer in modern medicine. Endometrial cancer is the most frequent type of gynecologic malignancy (its incidence is rapidly growing with the epidemic of obesity) - and unfortunately many patients suffer from late diagnosis and treatment in low volume centers - leading to decline in successful therapy. It is vital to qualify EC patients to proper treatment accordingly to new achievements in genetics.
- How original is the topic? What does it add to the subject area compared with other published material?
- The manuscript sums up the latest research and consensus of international societies; it adds important clinical insight and confront the guidelines with real life treatment options. It describes mode of action in many different types of EC in one place, so it may be very useful for clinicians.
- Is the text clear and easy to read?
- In my opinion it is well written and easy to read. It contains tables making it easy to find important information and there are many sub-paragraphs with titles making it easy to find important information .
- Are the conclusions consistent with the evidence and arguments presented?
- Yes. The reference list is extensive and up-to-date. The conclusions are drawn from multiple high-quality research and level of evidence is clearly presented after each statement.
- Do they address the main question posed?
- In my opinion publication of this manuscript may be very useful for clinicians and cited by many further researchers. Of course it is a review manuscript.
Author Response
Thank You for review and favorable assessment
Reviewer 3 Report
Well written review aiming at actualizing the Polish guidelines on endometrial carcinoma.
The authors should emphasize the relevance in the international field and strengths and limitations of their manuscript.
Author Response
The authors should emphasize the relevance in the international field and strengths and limitations of their manuscript.
Answer:
Relevance in the international field:
For the first time recommendations were developed according to standards set by the guideline tool AGREE II (Appraisal of Guidelines for Research and Evaluation) in Gynecologic Oncology.
The strengths of guidelines are comprehensiveness and up-to-date. We have included in one document the current evidence for diagnosis, treatment (including fertility-sparing, management of hereditary endometrial cancer syndromes), and follow-up of endometrial carcinoma.
The limitation is the uncertainty of how many of this guideline authors using AGREE II were methodological experts.
Reviewer 4 Report
With great interest, I read the manuscript on an important clinical issue that significantly impacts the clinical problems of many patients. The matter is certainly of the utmost importance, affecting large groups of women around the world. Nevertheless, the authors did not shy away from minor shortcomings and omissions, which I will present in points:
-
The text definitely needs linguistic correction: for example, even in the abstract there are spelling and punctuation errors already: "sequencies"->sequences, or multiple missing: "the", "an", hyphens.
-
The authors cite guidelines from AOTMIT, which is a government agency and the guidelines have not been published in the scientific literature. In the references, the authors provide a link to the guidelines - the link does not work. First of all, you should avoid providing this type of link, because as you can see by the time the paper is submitted for review it already becomes outdated - please provide a permanent electronic ID! Given that the guidelines implement EUnetHTA recommendations, it seems reasonable to refer to the source guidelines. In addition, the classification of scientific reports contained in the AOTMIT guidelines is a modification of the classification given by "Undertaking systemic reviews of research on effectiveness: CRD guidelines for those carrying out or commissioning reviews. CRD report #4, University of York, York 1996.", which are generally accepted principles for the scientific evaluation of evidence, and one would rather refer to the original guidelines.
-
The acronym AOTMIT is developed in some places incorrectly as "The Agency for Health Technology Assessment and Tariffication System." while in other places correctly: The Agency for Health Technology Assessment and Tariff System. Please standardize this.
-
Table 3:
-the "the" is missing from the title before "endometrium"
- "oryginal" in the heading is not an English word
-
p5v83
The categorization should be named as “Bokhman's categorization” of endometrial carcinomas when it first appears in the article. The Type 1 carcinomas are of endometrioid type, arise in younger perimenopausal patients in a background of endometrial hyperplasia with unopposed oestrogen stimulation, and are usually indolent in their behaviour.
Type 2 carcinomas, on the other hand, are prototypically of serous type, tend not to be associated with unopposed oestrogen stimulation, arise in a background of atrophic endometrium in older postmenopausal patients and
behave aggressively. It should be stated that in addition to endometrioid adenocarcinomas, Type 1 carcinomas include mucinous subtype of endometrial adenocarcinoma! -
In Section 3.1, the authors aptly refer to the indications for endometrial biopsy in postmenopausal bleeding patients and patients on tamoxifen therapy.
It should be noted that American guidelines in the case of persistent or recurrent uterine bleeding should prompt a histologic evaluation of the endometrium regardless of endometrial thickness PMID 25798986. If the authors of the recommendation think otherwise, then somehow the ACOG guidelines should be referred to.
The information on the necessity of biopsy in patients previously diagnosed with endometrial hyperplasia needs to be supplemented, as this is a risk group for endometrial cancer.
There is also a lack of information on the indications for endometrial biopsy in menstruating patients - who may also develop endometrial cancer.
The experts' position that there is no preferred way to obtain material from the uterine cavity is not fully understood. Even if the authors believe that all biopsy methods are equally valuable, they need to refer to and discuss the numerous positions and meta-analyses favoring hysteroscopy (higher accuracy and superior diagnostic yield) : PMID: 31482553, 30060741, 11846711, 23090535, 25798986, 11042572.
-
Reference [39] is outdated and should be replaced with PMID 30379327.
This paper [39] does not refer to robotic surgery so no conclusions can be made regarding the superiority of TRH over laparotomy. The conclusion should be revised (in the TRH section) or the claim should be supported by an additional reference.
Author Response
- The text definitely needs linguistic correction: for example, even in the abstract there are spelling and punctuation errors already: "sequencies"->sequences, or multiple missing: "the", "an", hyphens.
Answer: it was corrected- see revised manuscript
- The authors cite guidelines from AOTMIT, which is a government agency and the guidelines have not been published in the scientific literature. In the references, the authors provide a link to the guidelines - the link does not work. First of all, you should avoid providing this type of link, because as you can see by the time the paper is submitted for review it already becomes outdated - please provide a permanent electronic ID! Given that the guidelines implement EUnetHTA recommendations, it seems reasonable to refer to the source guidelines.
Answer: Thank You for this suggestion. Permanent electronic link was added
In addition, the classification of scientific reports contained in the AOTMIT guidelines is a modification of the classification given by "Undertaking systemic reviews of research on effectiveness: CRD guidelines for those carrying out or commissioning reviews. CRD report #4, University of York, York 1996.", which are generally accepted principles for the scientific evaluation of evidence, and one would rather refer to the original guidelines.
Answer: Indeed, this publication serves as instruction on how to provide meta-analyses of randomized trials. The hierarchy of study designs, described on page 23 in “CRD guidelines for those carrying out or commissioning reviews”, is close to our table. Unfortunately, we have to stay with polish HTA agency recommendations (polish table) as we work with them during the assessment of scientific reports
- The acronym AOTMIT is developed in some places incorrectly as "The Agency for Health Technology Assessment and Tariffication System." while in other places correctly: The Agency for Health Technology Assessment and Tariff System. Please standardize this.
Answer: acronym was standardized: The Agency for Health Technology Assessment and Tariffication System.
- Table 3:
-the "the" is missing from the title before "endometrium"
- "oryginal" in the heading is not an English word
Answer: Both corrected
- p5v83
The categorization should be named as “Bokhman's categorization” of endometrial carcinomas when it first appears in the article. The Type 1 carcinomas are of endometrioid type, arise in younger perimenopausal patients in a background of endometrial hyperplasia with unopposed oestrogen stimulation, and are usually indolent in their behaviour. Type 2 carcinomas, on the other hand, are prototypically of serous type, tend not to be associated with unopposed oestrogen stimulation, arise in a background of atrophic endometrium in older postmenopausal patients and behave aggressively. It should be stated that in addition to endometrioid adenocarcinomas, Type 1 carcinomas include mucinous subtype of endometrial adenocarcinoma!
Answer: It has been changed in agreement with reviewer suggestion
(...) In terms of pathogenesis, endometrial cancer was categorized by Bokhman into two subgroups: (…)
We don’t describe Bokhman types in the way suggested by the reviewer because we would like to focus on modern molecular classification.
- In Section 3.1, the authors aptly refer to the indications for endometrial biopsy in postmenopausal bleeding patients and patients on tamoxifen therapy.
It should be noted that American guidelines in the case of persistent or recurrent uterine bleeding should prompt a histologic evaluation of the endometrium regardless of endometrial thickness PMID 25798986. If the authors of the recommendation think otherwise, then somehow the ACOG guidelines should be referred to.
Answer:
Utilizing the tool AGREE II requires asking the question and answering by analyzing available scientific evidence. Section 3.1 is about the indication for biopsy.
We have asked in which clinical situation pathological assessment of endometrium has acceptable sensitivity and specificity.
The answer was as described in our recomendtaions.
ACOG guidelines has not been consistant with AGREE II thus contains questionable recommendations like this mentioned by the reviewer: (…) in the case of persistent or recurrent uterine bleeding should prompt a histologic evaluation of the endometrium regardless of endometrial thickness (…) It suggests that ECU should be performed for eg. in women with submucosal fibroids or in women with abnormal uterine bleeding (AUB) complication of anticoagulant therapy.
How safely sample sonographically invisible endometrium? This ACOG recommendation could generate an increased risk of uterine injury and unjustified increase in costs.
The information on the necessity of biopsy in patients previously diagnosed with endometrial hyperplasia needs to be supplemented, as this is a risk group for endometrial cancer.
Answer:
The management of endometrial hyperplasia is a separate subject thus it was not included into consideration during preaparing endometrial cancer guidelines. Endometrial biopsy after hormonal therapy of woman with simple hyperplasia is an integral part of treatment.
There is also a lack of information on the indications for endometrial biopsy in menstruating patients - who may also develop endometrial cancer.
Answer:
This was explained in recomendation regarding asymptomatic women (pre- and post-menopausal) - see text below
(...) The endometrial thickness cut-off at which biopsy in asymptomatic women (pre- and post-menopausal) in general population and in those with increased risk of endometrial cancer (PCOS, obesity, no childbirths, late menopause), which would have acceptable sensitivity and specificity has not been established. Thus, individual approach is recommended [expert opinion] [Strength of evidence V] (grade of recommendation 2B) (…)
The experts' position that there is no preferred way to obtain material from the uterine cavity is not fully understood. Even if the authors believe that all biopsy methods are equally valuable, they need to refer to and discuss the numerous positions and meta-analyses favoring hysteroscopy (higher accuracy and superior diagnostic yield) : PMID: 31482553, 30060741, 11846711, 23090535, 25798986, 11042572.
Answer
We do not put hysteroscopic biopsy into consideration because it was suggested that hysteroscopy in patients with endometrial cancer hints a risk for cancer cell dissemination within the peritoneal cavity:
[Polyzos NP, Mauri D, Tsioras S, Messini CI, Valachis A, Messinis IE. Intraperitoneal dissemination of endometrial cancer cells after hysteroscopy: a systematic review and meta-analysis. Int J Gynecol Cancer. 2010 Feb;20(2):261-7. doi: 10.1111/igc.0b013e3181ca2290. PMID: 20169669.
Very recent study also confirmed increased risk of intraperitoneal dissemination of malignant cells, but suggesting that this may be associated with intrauterine pressure >80 mmHg but not with stages I-II. [ Dong H, Wang Y, Zhang M, Sun M, Yue Y. Whether preoperative hysteroscopy increases the dissemination of endometrial cancer cells: A systematic review and meta-analysis. J Obstet Gynaecol Res. 2021 Sep;47(9):2969-2977. doi: 10.1111/jog.14897. Epub 2021 Jun 21. PMID: 34155733]
Thus our guidelines contains data on sensitivity, false negative rate and adecuation for abrasion and aspiration biopsy only.
Suggested by the reviewer literature potentially favorizing hysteroscopy is described below:
- PMID: 31482553 – this is not metaanalysis but guidelines only ( strength of evidence V)
- PIMID: 30060741 – this is systemic review with metanalysis supporting the use of Pipelle in the management of low-risk women presenting in the outpatient setting with symptomatic AUB when combined with clinical assessment and ultrasound scanning. Does not refer to high risk endometrial cancer cases
- PIMID 11846711 – Outdated (2001) observational study with control group – low risk patients – strength of evidence IV
- PIMID 23090535, - review from 2012 – strength of evidence V
- PIMID 25798986 - Practice Bulletin No. 149: Endometrial cancer - strength of evidence V
- PIMID 11042572 – included in our references: number [19]
- Reference [39] is outdated and should be replaced with PMID 30379327.
This paper [39] does not refer to robotic surgery so no conclusions can be made regarding the superiority of TRH over laparotomy. The conclusion should be revised (in the TRH section) or the claim should be supported by an additional reference.
Answer: Indeed we have missed this update from 2018
Reference [39] was changed as well as additional reference describing role of TRH was added [new 40]
Current text of recomendations as follows
As minimally invasive surgery (total laparoscopic hysterectomy /TLH/, and total robotic hysterectomy /TRH/) does not compromise the prognosis and has a significant advantage in perioperative and postoperative outcomes over open surgery [39, 40] [Strength of evidence III A, IIIA], is recommended where possible (grade of recommendation 2A).
[40] Dinoi G, Ghoniem K, Murad MH, Segarra-Vidal B, Zanfagnin V, Coronado PJ, Kyrgiou M, Perrone AM, Zola P, Weaver A, McGree M, Fanfani F, Scambia G, Ramirez PT, Mariani A. Minimally Invasive Compared With Open Surgery in High-Risk Endometrial Cancer: A Systematic Review and Meta-analysis. Obstet Gynecol. 2023 Jan 1;141(1):59-68. doi: 10.1097/AOG.0000000000004995.
Round 2
Reviewer 4 Report
1. Thank you for addressing my comments on the consistency of recommendations with major medical societies (e.g. ACOG), comments on hysteroscopic endometrial biopsy and indications for biopsy. The reviewer accepts the responses, although he has a different opinion on the subject. However, each scientific society has the right to issue medical recommendations on its own behalf and at its own responsibility. It would be good if at least some of the explanations from the response to the reviewer were reflected in the text of the paper so that it is unambiguous and not confusing to the uninitiated reader. For example, it would be helpful to clearly emphasize that the management of endometrial hyperplasia (which is a precancerous condition) is a separate subject not covered by the recommendation. That is where my role as a reviewer ends.
2. The correct name of the agency is: The Agency for Health Technology Assessment and Tariff (sic!) System. Please make a proper standardization.
Author Response
Thank you for the discerning review. It was a great pleasure to work with your comments.
The suggestion about hyperplasia was added. See the current version below.
"(...) The endometrial thickness cut-off at which biopsy in asymptomatic women in the general population (pre- and postmenopausal) and in those with increased risk of endometrial cancer (PCOS, obesity, no childbirths, late menopause)*, which would have acceptable sensitivity and specificity has not been established. Thus, an individual approach is recommended [expert opinion] [Strength of evidence V] (grade of recommendation 2B).
*Note: the management of endometrial hyperplasia (a precancerous condition) is a separate subject not covered by this recommendation. (...)"